# Helical jackknives control the gates of the double-pore K⁺ uptake system KtrAB

**Marina Diskowski[1], Ahmad Reza Mehdipour[2], Dorith Wunnicke[1], Deryck J Mills[3], Vedrana Mikusevic[1], Natalie Bärland[1,3], Jan Hoffmann[4], Nina Morgner[4], Heinz-Jürgen Steinhoff[5], Gerhard Hummer[2,6], Janet Vonck[3]\*, Inga Hänelt[1]\***

[1]Institute of Biochemistry, Goethe-University, Frankfurt, Germany; [2]Department of Theoretical Biophysics, Max Planck Institute of Biophysics, Frankfurt, Germany; [3]Department of Structural Biology, Max Planck Institute of Biophysics, Frankfurt, Germany; [4]Institute for Physical and Theoretical Chemistry, Goethe-University, Frankfurt, Germany; [5]Department of Physics, University of Osnabrück, Osnabrück, Germany; [6]Institute of Biophysics, Goethe-University, Frankfurt, Germany

**\*For correspondence:** janet. vonck@biophys.mpg.de (JV); haenelt@biochem.uni-frankfurt.de (IH)

**Competing interests:** The authors declare that no competing interests exist.

**Abstract** Ion channel gating is essential for cellular homeostasis and is tightly controlled. In some eukaryotic and most bacterial ligand-gated K⁺ channels, RCK domains regulate ion fluxes. Until now, a single regulatory mechanism has been proposed for all RCK-regulated channels, involving signal transduction from the RCK domain to the gating area. Here, we present an inactive ADP-bound structure of KtrAB from *Vibrio alginolyticus*, determined by cryo-electron microscopy, which, combined with EPR spectroscopy and molecular dynamics simulations, uncovers a novel regulatory mechanism for ligand-induced action at a distance. Exchange of activating ATP to inactivating ADP triggers short helical segments in the K⁺-translocating KtrB dimer to organize into two long helices that penetrate deeply into the regulatory RCK domains, thus connecting nucleotide-binding sites and ion gates. As KtrAB and its homolog TrkAH have been implicated as bacterial pathogenicity factors, the discovery of this functionally relevant inactive conformation may advance structure-guided drug development.

## Introduction

Potassium ions (K⁺) are the main cations in living cells from all kingdoms of life. Within the cell, the K⁺ concentration is responsible for a plethora of tasks, including pH homeostasis (*Bakker and Mangerich, 1981*; *Kroll and Booth, 1981*; *Plack and Rosen, 1980*), cell growth, maintenance of osmolarity and cell volume (*Schultz et al., 1963*), movement and electrical signaling (*Hille, 2001*). Consequently, K⁺ translocation across the membrane is a highly regulated process, facilitated by integral membrane proteins.

The majority of the prokaryotic ligand-gated K⁺ channels and many eukaryotic systems consist of a single tetrameric pore regulated by covalently linked cytoplasmic ligand-binding domains, termed regulator of K⁺ conductance (RCK) (*Jiang et al., 2001*, *2002b*). Based on the structural knowledge of various K⁺ uptake systems, a three-stage regulatory mechanism via the RCK domains has been proposed (*Chakrapani and Perozo, 2007*; *Hite et al., 2015*; *Jiang et al., 2002a*, *2002b*; *Ye et al., 2006*; *Yuan et al., 2011*). Initially, binding of a ligand to the RCK domains induces a conformational change. This rearrangement triggers the movement of flexible linkers that connect the RCK domains with the C-termini of the pore domains. The third step is initialized by the altered flexibility of the linkers, which translates the mechanical signal into an opening of the gating region, the so-called helical bundle. Finally, the open ion pathway allows free flow of K⁺.

In contrast to the aforementioned K⁺ channels, the pore subunits (KtrB/TrkH) of the osmoprotective systems Ktr and Trk merely interact with the cytoplasmic RCK subunits (KtrA/TrkA), missing the covalent helical linkage (Cao et al., 2013; Vieira-Pires et al., 2013). Furthermore, instead of a single pore, two neighboring pores are regulated by one octameric ring of RCK subunits (Albright et al., 2006). Hence, an adapted regulatory mechanism is required taking these distinct structural features of the Ktr and Trk systems into consideration (Albright et al., 2007; Cao et al., 2011).

Ktr and Trk systems exhibit a very similar structure. Each KtrB and TrkH monomer consists of four pairs of covalently linked transmembrane helices (D1-D4) that surround the ion pathway and

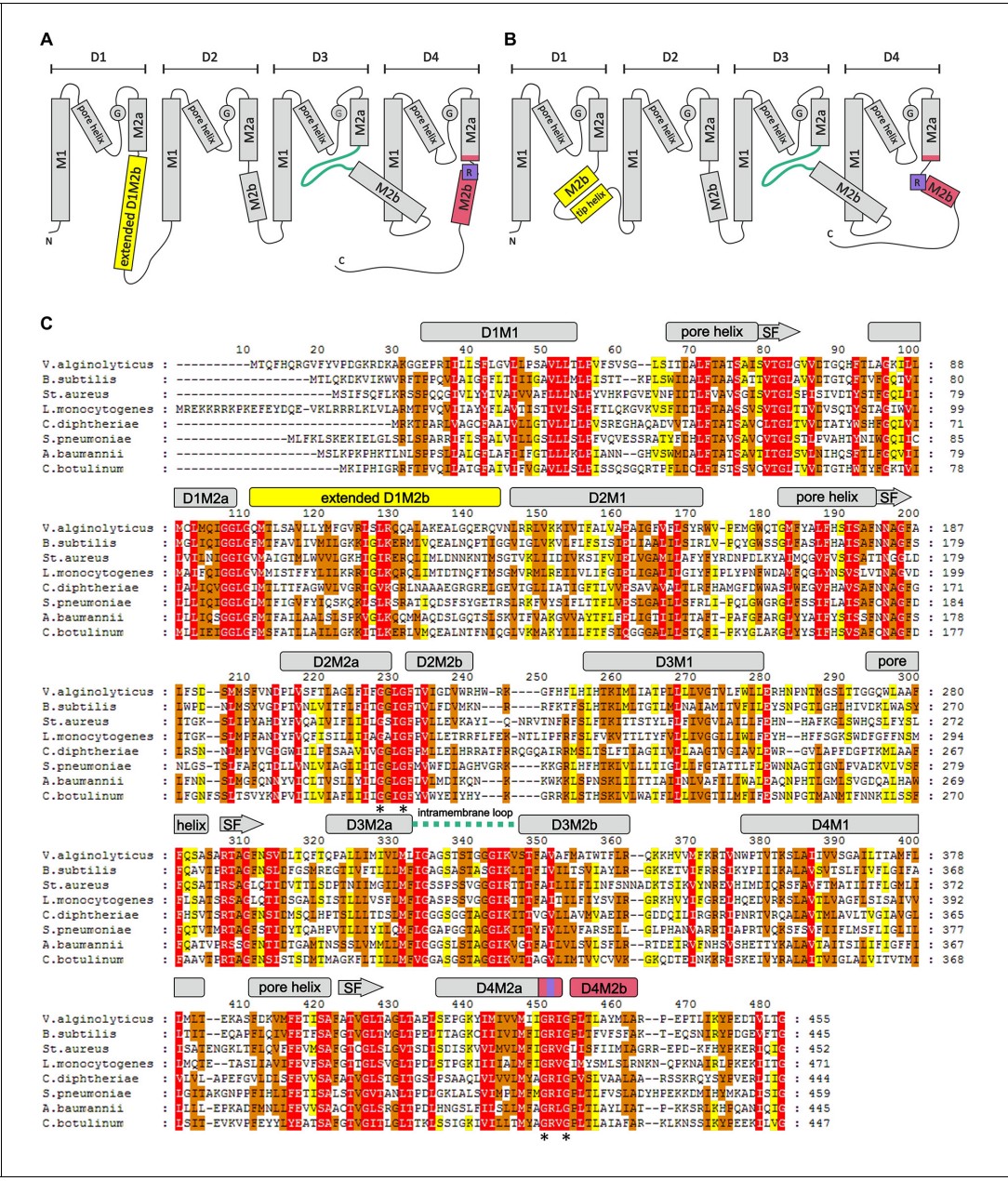

**Figure 1.** Overview of the KtrB subunit. Topology diagram of KtrB subunit structure in the presence of ADP (**A**) and ATP (**B**). (**C**) Amino acid sequence alignment of different KtrBs using Clustal Omega and GeneDoc and secondary structure of VaKtrB in the presence of ADP. α-helices and β-strands are shown as rectangles and arrows, respectively. Sequence conservation color-coded: White on red, 100%; black on orange, ≥80%; black on yellow, ≥60%. SF = selectivity filter. Discussed motifs are highlighted: intramembrane loop is colored in green, D1M2b helix and tip helix in yellow, D4M2b helix in magenta and the conserved arginine in purple. Asterisks (*) mark conserved gating hinge glycine residues.

together constitute the pore (*Figure 1A,B*). The two transmembrane helices M1 and M2 of each repeat are joined by pore loops (P loops), which form the selectivity filter (*Cao et al., 2013*; *Durell and Guy, 1999*; *Vieira-Pires et al., 2013*). Compared to the TVGYG motif of classical $K^+$ channels, the selectivity filter of KtrB and TrkH is less conserved (*Zhou et al., 2001*); a single glycine residue of each of the four P loops converges within the ion pathway (*Figure 1*) (*Tholema et al., 1999*, *2005*). An intramembrane loop formed by the central part of broken D3M2 directly beneath the selectivity filter represents a unique structural motif that regulates $K^+$ flow by blocking and opening of the pore. The intramembrane loop (*Hänelt et al., 2010a*, *2010b*) together with a highly conserved arginine residue in D4M2 (*Cao et al., 2011*) forms the molecular gate (cf. *Figure 1C*). Regulation of the pore subunits, KtrB and TrkH, is thought to occur via nucleotide-dependent conformational changes of the octameric RCK subunits (*Cao et al., 2013*; *Kröning et al., 2007*; *Szollosi et al., 2016*; *Vieira-Pires et al., 2013*). X-ray structures revealed distinct ATP- or ADP-induced conformational changes within the RCK ring, whereas the pore conformation was almost identical (*Szollosi et al., 2016*; *Vieira-Pires et al., 2013*). However, the link between conformational changes within the RCK subunits and pore permeability remained elusive.

Here, we present a cryo-electron microscopy (cryo-EM)-based structural model of ADP-bound KtrAB, combined with pulsed electron paramagnetic resonance (EPR) measurements and molecular dynamics (MD) simulations. Our data reveal the direct regulation of the pore subunits by ligand-induced conformational changes within the octameric RCK ring. The proposed new regulatory mechanism for Ktr/Trk exerts control over the two neighboring pores and represents a surprising variation to the three-stage regulatory mechanism observed in single-pore RCK channels.

## Results

### Structure of the inactive KtrAB complex

To obtain a cryo-EM map of ADP-bound KtrAB, full-length KtrAB from *Vibrio alginolyticus* was heterologously produced in *Escherichia coli*. The detergent-solubilized complex used for cryo-EM data collection was shown to bind ADP and ATP with low micromolar affinities (*Figure 2*). A total of 20,500 particle images of ADP-bound KtrAB were combined to generate a cryo-EM map at an overall resolution of 6.6 Å with D2 symmetry imposed (*Figures 3A* and *4*). The density map displays an octameric ring of KtrAs, associated on both sides with a KtrB dimer in a detergent belt (*Figure 3A, B*). This $KtrB_2A_8B_2$ composition was confirmed by negative stain EM of the detergent-solubilized complex and native mass spectrometry (*Figure 4A,B* and *Figure 4—figure supplement 1*). This assembly, also observed in crystal structures of KtrAB and TrkAH homologs (*Cao et al., 2013*; *Vieira-Pires et al., 2013*), is probably formed during protein purification due to the symmetry of the octameric KtrA ring, which has identical binding sites for the KtrB dimer on both sides. The C-terminal domains of KtrA, the so-called C lobes at the periphery of the KtrA ring, are poorly resolved, indicating an increased flexibility of these areas (*Figure 4F*). The KtrA ring has an oval shape (*Figure 5C*), as has been described for the homologous proteins within the complex (*Cao et al., 2013*), as well as for the solitary KtrA ring in the presence of ADP (*Albright et al., 2006*). A remarkable feature of the KtrAB map, which has not previously been observed, is an elongated density protruding from each KtrB monomer into the octameric KtrA ring (*Figure 3A,B*). In contrast, no densities are visible for D1M2b and the tip helix, which form a helix hairpin at the membrane surface in the ATP-bound structure (cf. *Figure 5A,B*), suggesting large nucleotide-dependent conformational changes in this region. We constructed a model of ADP-bound KtrAB by flexibly fitting a homology model of KtrAB of *V. alginolyticus* in the ATP-bound conformation (based on the corresponding X-ray structure of KtrAB from *Bacillus subtilis* [*Vieira-Pires et al., 2013*]) into the EM map (*Figure 3C*). The most prominent feature in the ADP-bound KtrAB model is a subunit-interconnecting helix, 58 Å in length, which spans the entire membrane and protrudes into the KtrA ring (*Figures 3C* and *5A*). This helix is an extension of helix D1M2, which in the ATP-bound structure is ~28 Å long. The adjacent C-terminal helix D4M2b is extended by 18 Å (3.3 helical turns) in the ADP-bound conformation compared to the ATP-bound state and also spans the entire membrane (cf. *Figure 5A,B*). These major conformational changes in D1M2 and D4M2 together obviously narrow the pores in the ADP-bound conformation (*Figure 5A,B*). Apart from these significant changes, the overall structure of the membrane part of the ADP-bound model resembles the corresponding

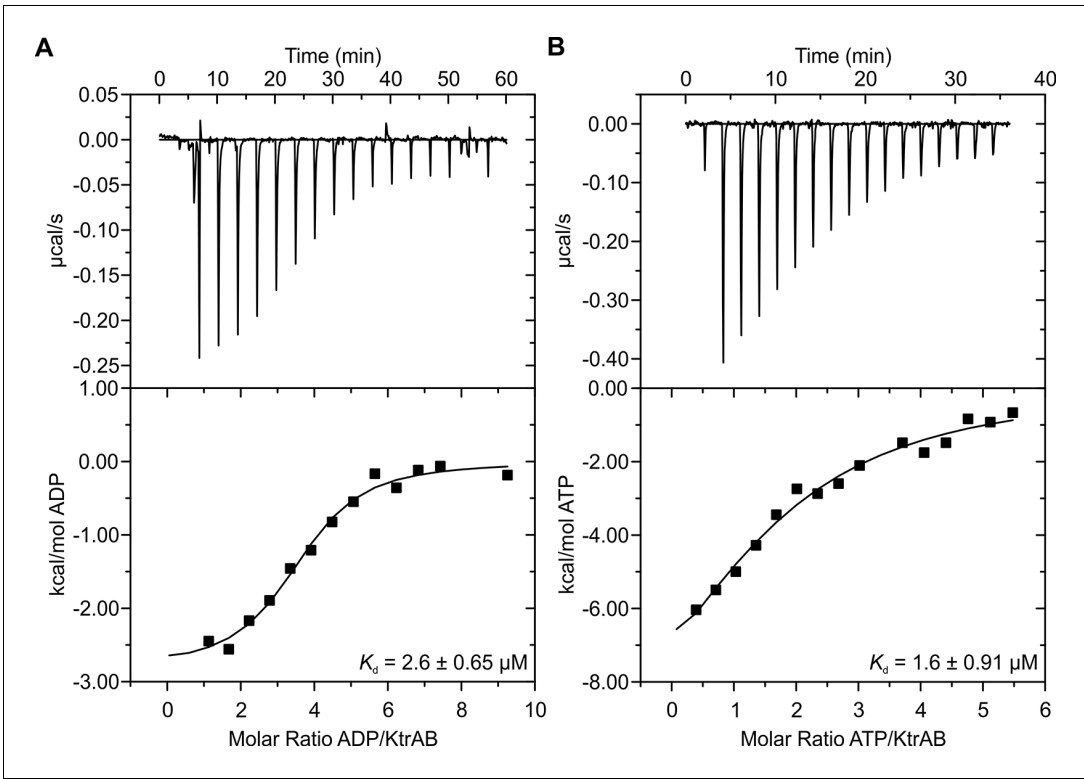

**Figure 2.** Binding of KtrAB to nucleotides examined by ITC measurements. Upper panels: raw heat exchange data, associated with (**A**) ADP or (**B**) ATP binding to detergent-solubilized KtrAB. Lower panel: integrated injection heat pulses, normalized per mole of injection, reveal differential binding curves calculated by a one-site binding model with determined $K_d$ values of 3 µM for ADP and 2 µM for ATP.

conformation of KtrAB in its ATP-bound form (cf. *Figure 6A,B*, *Video 1*). Similar to the ATP-bound conformation, the KtrB protomers interact with each other via their C-termini; each terminus extends into the cytoplasmic cavity of the neighboring protomer and in addition forms a lateral contact with the octameric ring of KtrA (*Figure 3C*).

## Helix D1M2 connects KtrB with KtrA

A comparison of the ADP- and ATP-bound structural models allows us to further analyze the newly identified coupling helix D1M2 that connects cytoplasmic KtrA with membrane-embedded KtrB and probably is a key feature for the regulation of Ktr. In the ADP-bound conformation, D1M2 forms one continuous, elongated α-helix that reaches into the octameric KtrA ring, probably enabling direct interactions with nearby residues close to the ADP-binding sites (*Figure 5C*). An overlay of the KtrB subunits in the ADP-bound state and the KtrA ring in its ATP-bound conformation implies that this extended α-helix is unlikely to form in the presence of ATP, due to steric hindrance (*Figure 5D*). In the crystallized ATP-bound conformation, D1M2 is twice broken, forming a short transmembrane α-helix with a C-terminal helix hairpin extending to the cytoplasmic membrane surface (*Figures 5B* and *6B*).

The ADP- and ATP-bound structural models (*Cao et al., 2013*; *Vieira-Pires et al., 2013*) reveal that ATP interacts with two different KtrAs via its γ-phosphate group, whereas ADP interaction occurs within one subunit. These disparate interactions of the nucleotides allow the diamond-shaped conformation of the RCK domain in the ADP-bound state in presence of the elongated D1M2 helix, whereas the square-shaped ATP-bound state excludes the straightening of the α-helix by steric hindrance (cf. *Figure 5C,D*).

In order to confirm the nucleotide-induced conformational rearrangement of the D1M2 region, we used pulsed EPR measurements on the detergent-solubilized, spin-labeled KtrAB complex.

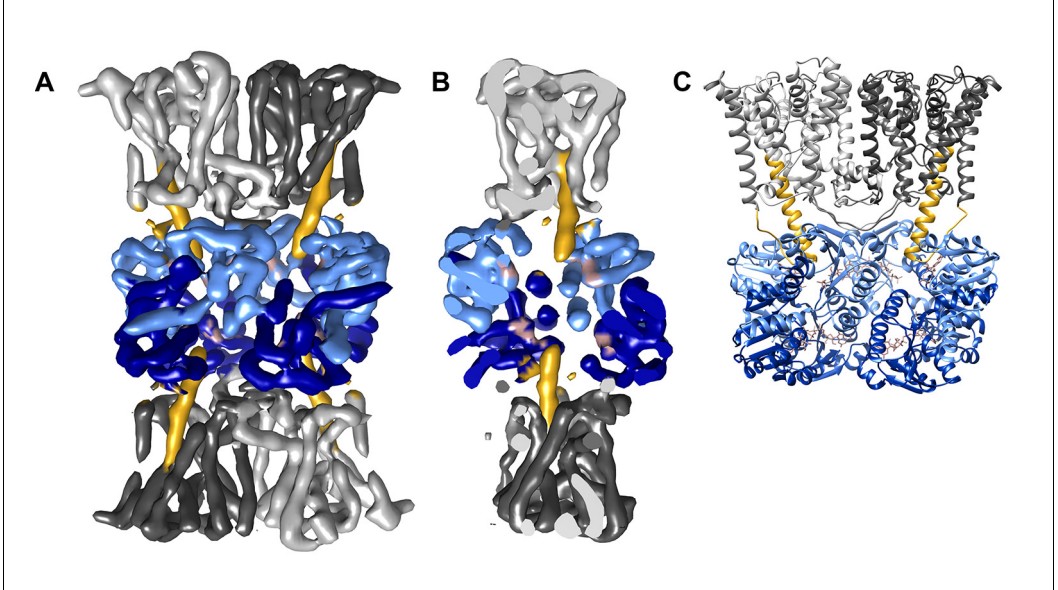

**Figure 3.** Three-dimensional reconstruction of ADP-bound KtrAB. (**A**) Side view of the cryo-EM map colored by protein subunits. (**B**) 90° rotated side view of the density map cut open to highlight the extended helices. (**C**) Structural model of ADP-bound KtrAB in the same orientation as in **A**. KtrBs, dark and light grey with helices D1M2, yellow; octameric ring of KtrA, dark and light blue; ADP in rosy brown.

Based on a MMM evaluation (*Polyhach et al., 2011*) of the different positions within our models' D1M2 region, we selected the position that potentially undergoes the largest nucleotide-regulated conformational changes. Thus, we introduced a single cysteine into a Cys-less KtrAB variant at position A122 in KtrB (*Figure 5—figure supplement 1A*). The functionality of the mutant was confirmed by a growth complementation assay at limiting K$^+$ concentrations (*Figure 5—figure supplement 1B*). Taking our model of the ADP-bound KtrB$_2$A$_8$B$_2$ complex into account, the presence of four KtrB protomers results in three distinct distance contributions. Therefore, a relatively broad total distance distribution in the range of 2 to 5 nm was predicted, reflecting the distances between four spin-labeled residues A122C within a single KtrB$_2$A$_8$B$_2$ complex (*Figure 5E*, dashed rosy brown line, *Figure 5—figure supplement 2*, upper panel). In comparison, the model of the ATP-bound KtrB$_2$A$_8$B$_2$ complex led us to predict a shift to longer interspin distances of 6 to 10 nm (*Figure 5E*, dashed blue line, *Figure 5—figure supplement 2*, lower panel). Indeed, the slow decline of the dipolar evolution functions in the presence of ATP indicates long distances, which according to the Tikhonov regularization center around 5.3 nm (*Figure 5E*, *Figure 5—figure supplement 3*). The shift to longer distances becomes more pronounced with increasing ATP concentrations. In analogy to the measurements in the presence of ATP, the experimentally determined dipolar evolution functions in the presence of ADP show a fast decay with poor oscillations, which correspond to broad distance distributions centered around 3.0 nm (*Figure 5E*, *Figure 5—figure supplement 3*). An increased ADP concentration results in a pronounced and fast decay, indicating an increased population at short distances. The distance distribution obtained in the presence of a high ADP concentration is in good agreement with the interspin distances predicted for the elongated helices. In both ATP conditions, the experimentally determined mean distances are shorter than the model-based, calculated data. However, the experimentally determined mean distances for the ATP-bound model are significantly longer compared to the model in the presence of ADP. The decrease in experimentally determined mean distance, compared to the one predicted for the ATP-bound model, is presumably caused by the limited dipolar evolution time (*de Vera et al., 2013*; *Jeschke et al., 2006*). The experimentally determined mean distances would likely increase with a prolonged d2 time. Additionally, the D1M2 region could of cause adopt a slightly different conformation than given in the ATP-bound model, which could result in slightly shorter distances. In summary, the measurements reflect the nucleotide-dependent conformational change of the D1M2 region between the formation of the helix

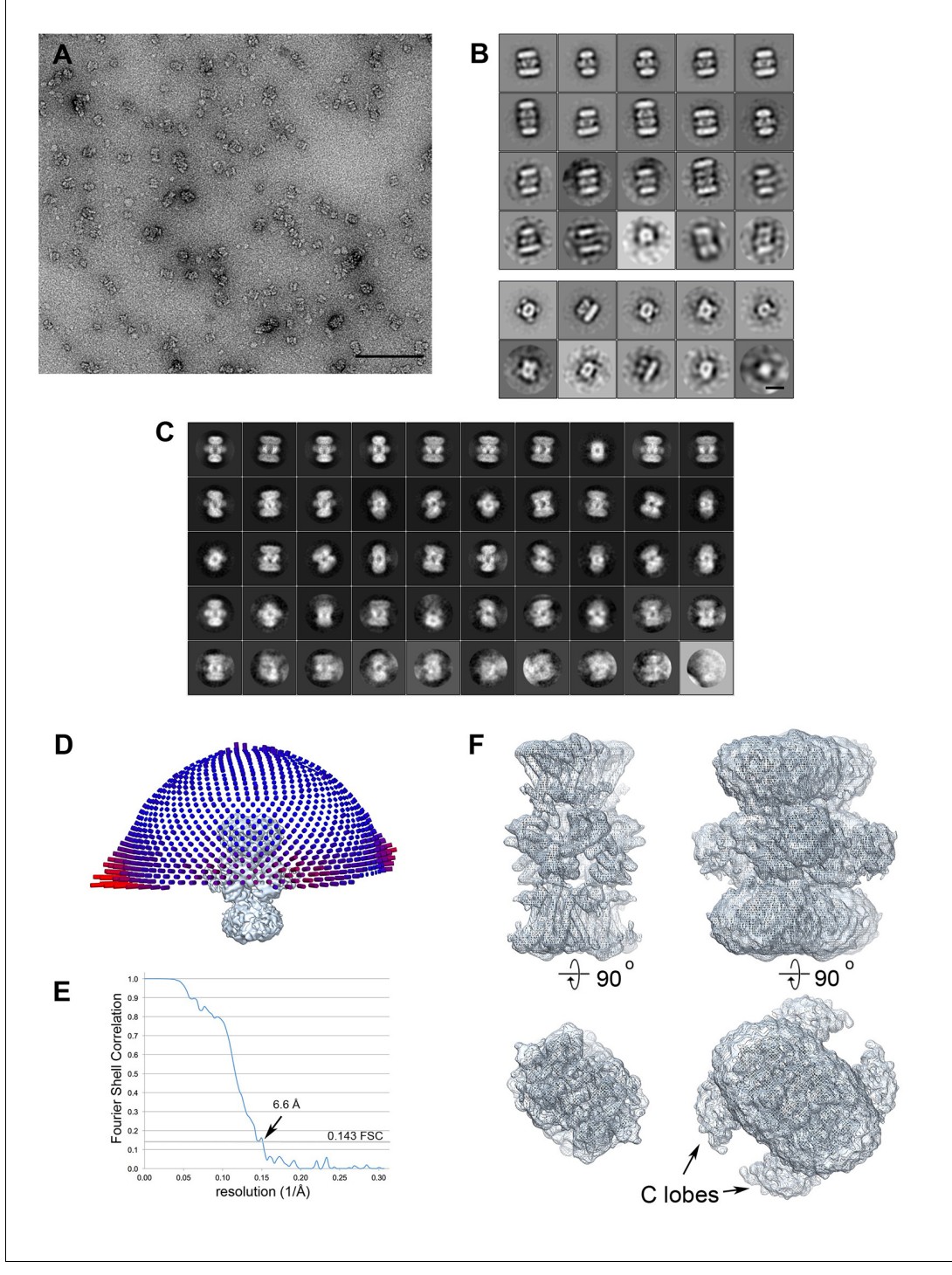

**Figure 4.** Electron microscopy data. (**A**) KtrAB negatively stained with uranyl acetate. The scale bar is 100 nm. (**B**) Class averages from negatively stained particles. The top panel shows classes from a data set of ~1600 large particles. Most classes show a $KtrB_2A_8B_2$ arrangement, a few are $KtrB_2A_8A_8B_2$ complexes. The lower panel shows a classification of ~500 small particles. Most classes show an elliptic top view with four KtrA C-terminal dimers; two classes show a side view of a KtrAB complex. Scale bar: 10 nm. (**C**) 2D class averages of the final cryo-EM data set of 20,500 particles. The classes show different views of the complex, many with high-resolution detail. (**D**) The final cryo-EM density map with the orientation distribution. The length of each cylinder represents the number of particles viewed from that direction. Although the particles have preferred orientations (long red cylinders), there are no missing orientations. (**E**) Fourier Shell Correlation curve indicating a resolution of 6.6 Å using the 0.143

*Figure 4 continued on next page*

*Figure 4 continued*

criterion (dotted line). (**F**) Top view and side view of the density map with high (left) and low (right) contour level. At low contour level, the detergent micelle around KtrB and the characteristic peripheral C lobes around the octameric KtrA ring are visible.

The following figure supplement is available for figure 4:

**Figure supplement 1.** Mass spectrum of the detergent-solubilized KtrAB complex.

hairpins (in the presence of ATP) and elongated helix (in the presence of ADP). In the absence of nucleotides, the dipolar evolution function represents an intermediate between the two functions in the presence of ADP and ATP, respectively. This results in a distance distribution between 2 and 6 nm, with an equal probability of contributions from both conformations (*Figure 5—figure supplement 3*). Taken together, the analysis of our EPR data reveals for the first time a significant nucleotide-dependent movement of the D1M2 region of the K$^+$-translocating subunit KtrB and suggests a concentration-dependent equilibrium between the two states.

## Nucleotide-dependent K$^+$ gating

Previous in vivo and in vitro studies have demonstrated the activating effect of ATP on Ktr and Trk systems, while ADP was hypothesized to act as an inactivating factor (*Cao et al., 2013*; *Kröning et al., 2007*; *Vieira-Pires et al., 2013*). Thus, nucleotide-induced conformational changes in KtrA should affect the gating region of KtrB. The molecular gate in KtrB is thought to be formed by the intramembrane loop located just below the selectivity filter and a highly conserved arginine residue (R427 in *Va*KtrAB) in D4M2 situated in the same plane (*Figure 6A,B* and *Figure 1C*) (*Cao et al., 2011*, *2013*; *Hänelt et al., 2010a*, *2010b*). Interestingly, also the kinks in α-helices D1M2 and D4M2 introduced by the conformational change from the ADP- to the ATP-bound state are located in this region (*Figure 6A,B*). The kink in D4M2 may even directly reorient the gating arginine residue (R427) and move it away from the intramembrane loop as it is contained within the altered part of D4M2. Remarkably, the proposed gating region is also marked by a glycine-rich ring (*Figure 1C*, *Figure 6—figure supplement 1*), which in other K$^+$ channels is linked to the gating region and is referred to as glycine hinge (*Jiang et al., 2002a*). The essential aspect for the facilitation of the K$^+$ flux is the accessibility and the opening of the gate, which we have previously described as the movement of the intramembrane loop (*Hänelt et al., 2010a*, *2010b*). However, the resolution of our density map is not high enough to draw detailed conclusions on the conformation of the gate. Instead, MD simulations demonstrate that in the ADP-bound conformation, α-helices D1M2 and D4M2 interact with the gating motifs in KtrB and hinder water accessibility. In the presence of ATP, the number of water molecules in the cavity below the gating region increases due to the discontinuous structure of D1M2 and D4M2, making the gate accessible to water from the cytoplasmic side (*Figure 6C*, *Figure 6—figure supplement 2*). To explore the role of the intramembrane loop in water accessibility, we performed additional simulations of 10 ADP- and ATP-bound structural models each, in which this loop was modeled blindly (see Materials and methods). Results for the water accessibility in the ADP- and ATP-bound states agree with those of the original simulations. Within the model uncertainties, the conformation of the loop thus has a minimal effect on the hydration level of the gating region (*Figure 6C*). Therefore, the water accessibility of the gating region is mainly dependent on the D1M2 and D4M2 conformations.

For K$^+$ translocation, the increased water accessibility should be accompanied by the opening of the molecular gate. To directly determine nucleotide-dependent conformational changes in the gating region, we performed pulsed EPR measurements on KtrB and KtrAB variants with spin-labeled intramembrane loop residue T318CR1 (*Figure 5—figure supplement 1A*. The mutants' functionality again was confirmed using the complementation assay (*Figure 5—figure supplement 1B*). Since the movement of the intramembrane loop in KtrB alone has been previously shown to depend on a K$^+$ gradient and/or the lipid environment (*Hänelt et al., 2010b*), spin-labeled variants were initially reconstituted into liposomes. In the absence of the KtrA subunits, the intermolecular distances between residues T318CR1 in the KtrB dimer display a broad distance distribution both in the absence and presence of the K$^+$ gradient (*Figure 6—figure supplement 3A*), indicating the high

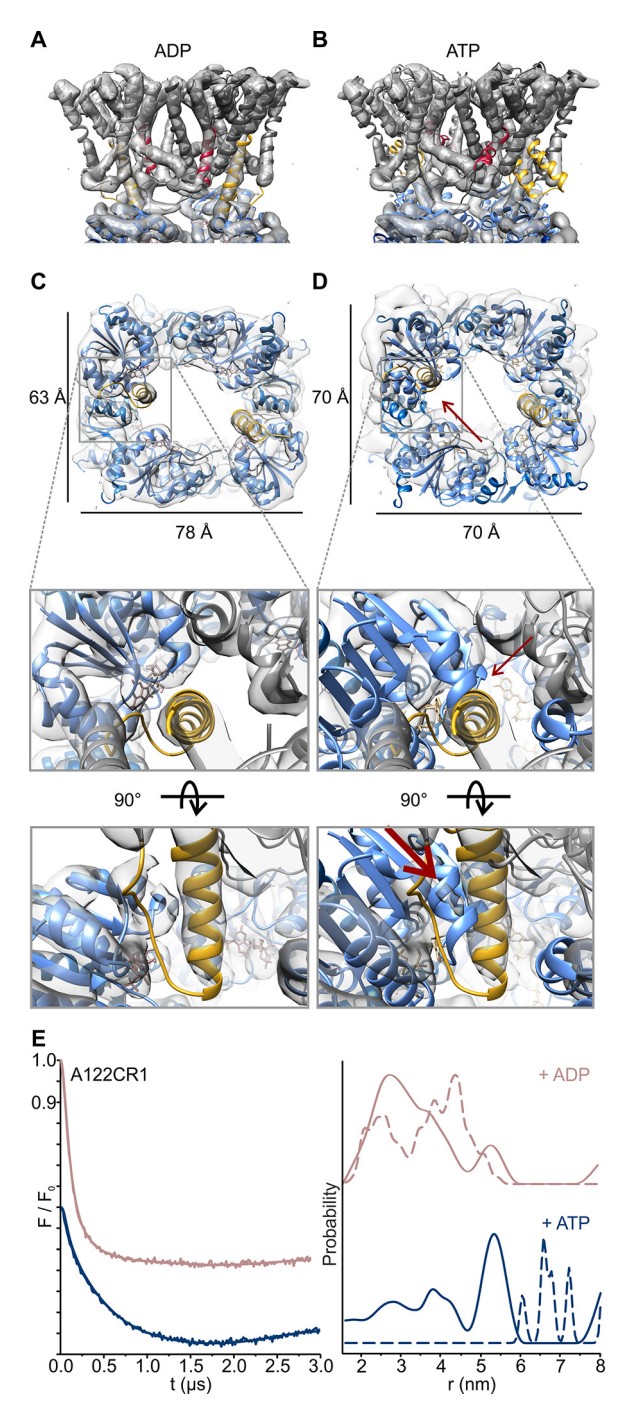

**Figure 5.** Nucleotide-induced conformational changes of the KtrAB complex. Overlay of the KtrAB density map in the presence of ADP with KtrAB model in the presence of (**A**) ADP and (**B**) ATP. KtrB subunits are depicted in light and dark grey, D1M2b helices in yellow, D4M2 helices in magenta, and RCK domains in blue. (**C**), (**D**) Top and side views of overlays of the density map with the ADP-bound KtrB model and the ADP- (**C**) or the ATP-bound (**D**) model of KtrA. KtrBs, grey and light grey with helices D1M2b, yellow; octameric ring of KtrA, blue and light blue; ADP, rosy brown; ATP, tan. Red arrows indicate steric hindrance of helix D1M2 with the ATP-bound KtrA. (**E**) Pulsed EPR data of spin-labeled variant KtrAB A122CR1 in detergent solution recorded at Q band (34 GHz). Dipolar evolution functions and the corresponding interprotomer distances obtained by Tikhonov regularization were determined in the presence of ADP (rosy brown) and ATP (blue), respectively. Dashed lines represent the

*Figure 5 continued on next page*

*Figure 5 continued*

predicted distance distributions of the ADP- and ATP- bound models, respectively, using the rotamer library analysis (*Polyhach et al., 2011*).

The following figure supplements are available for figure 5:

**Figure supplement 1.** Functionality of protein variants.

**Figure supplement 2.** Inter-protomer distances within the detergent-solubilized KtrB$_2$A$_8$B$_2$ complex predicted by rotamer library analysis.

**Figure supplement 3.** Pulsed EPR data of spin-labeled variant KtrAB A122CR1 in detergent solution recorded at Q band (34 GHz).

flexibility of the intramembrane loop independent of a K$^+$ gradient. The addition of external K$^+$ induces no significant change of the interspin distance distribution. In contrast, in the presence of KtrA and upon the inclusion of ADP inside the liposomal lumen, the recorded pulsed EPR spectrum shows a defined oscillation, which marks a narrow distance distribution centered around 3.3 nm. This narrow distance distribution reflects the immobilization of the intramembrane loop (*Figure 6D*, *Figure 6—figure supplement 3B*). However, in the presence of ATP inside the liposomes and upon the external addition of K$^+$, the oscillation of the spectrum becomes less defined, demonstrating a broader distance distribution and thus the movement of the intramembrane loop. Furthermore, the mean distance is shifted toward longer distances, suggesting an increased probability of an open conformation for facilitation of the flux of potassium ions (*Figure 6D*, *Figure 6—figure supplement 3B*). To examine the effect of nucleotides on the gating region only, we repeated the measurements on spin-labeled KtrAB variant T318C in a detergent environment (*Figure 6—figure supplement 3C, D*). Two different d2 times of 2 and 3 μs, respectively, were recorded to appropriately determine short and long distances. A narrow distance distribution of 3.5 nm was observed in the presence of ADP, representing a similar but slightly shifted mean distance distribution compared to the proteoliposome measurements. In the presence of ATP, the defined oscillation in the dipolar evolution function diminishes and the corresponding distance distribution significantly broadens. In comparison to the ATP-bound measurements in proteoliposomes the distance distributions are even broader, indicating an increased flexibility of the intramembrane loop in the detergent-solubilized sample. In summary, this set of measurements demonstrates that in the ADP-bound conformation of the KtrAB complex with its elongated α-helices D1M2 and D4M2, the molecular gate is stabilized in its inactivated, closed conformation. In contrast, the binding of ATP likely unlocks the intramembrane loop and enables the stochastic movement of the gate. In fact, MD simulations on the KtrB subunit of the ATP-bound X-ray structure of KtrAB from *B. subtilis* (*Vieira-Pires et al., 2013*) confirm the flexibility of the intramembrane loop in this conformation (*Figure 6—figure supplement 4*). Conformational changes from a closed state, which blocks the pore, to an open state, which allows water to access the selectivity filter, are enabled (*Figure 6—figure supplement 2*). Finally, as supported by the comparison of samples in detergent solution and liposomes, a K$^+$ gradient, the surrounding membrane and/or other factors yet to be identified are needed to stabilize the active, open conformation. Further structural investigations are required to resolve a molecular model of an open, active state.

## Discussion

This study presents, to the best of our knowledge, the first data on how gating in KtrB, and by extrapolation in TrkH, is regulated by nucleotide binding to the cytoplasmic RCK ring. Our results lead us to the proposal of an activation mechanism: Under non-activating conditions, ADP is bound to the RCK subunits in a conformation with extended α-helices D1M2, which are stabilized by specific, not yet identified interactions between KtrA and KtrB similar to a spring tension state. At the gating regions of KtrB α-helices D1M2 and D4M2 probably form stable interactions with the conserved arginine residue and/or residues of the intramembrane loop. These interactions lock the gate in its rigid, closed conformation. Upon a hyperosmotic shock the internal ATP concentration

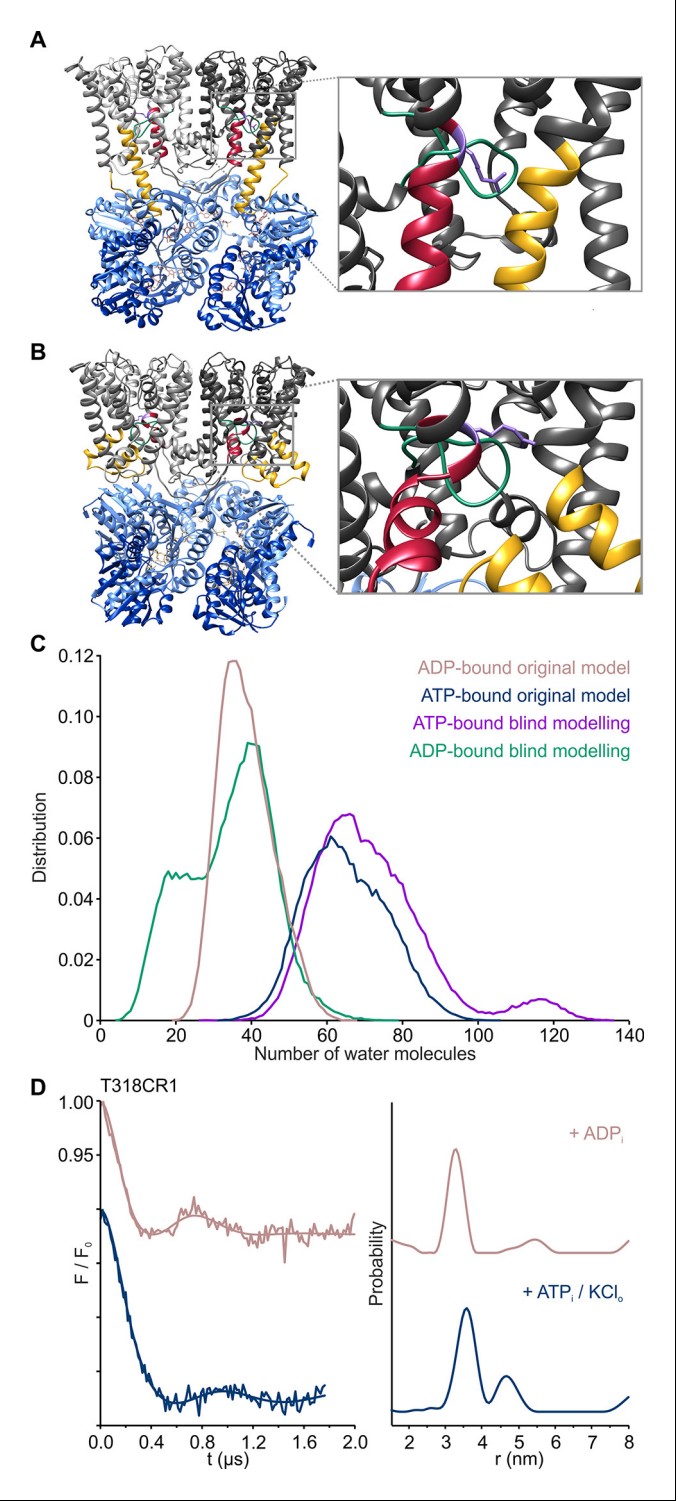

**Figure 6.** Gating mechanism of the KtrAB complex. (**A**), (**B**) Side views of one gating region of the KtrAB complex in the ADP- (**A**) and ATP-bound (**B**) conformation with helices at the front removed. KtrBs, grey and light grey with helices D1M2b, yellow and D4M2b, magenta; octameric ring of KtrA, blue and light blue; intramembrane loop, green; R427, purple. (**C**) Number of water molecules in the cavity below the gating region during MD simulations in the ADP-bound (rosy brown; simulation code: $ABDG_{VA}+ABDN_{VA}$) and ATP-bound states (blue; simulation code: $ABTG_{VA}+ABTN_{VA}$). For reference, averaged hydration numbers are shown from 10 additional simulations each in the ADP-bound (green; simulation code: $ABDGB_{VA}$) and ATP-bound states (purple; simulation code: $ABTGB_{VA}$), starting from structures in which the intramembrane loop was modeled blindly. The lower end of the cavity volume

*Figure 6 continued on next page*

*Figure 6 continued*

was consistently defined by a plane passing through the lipid head groups. (**D**) Pulsed EPR data of spin-labeled variant KtrAB T318CR1 reconstituted in liposomes recorded at Q band (34 GHz). Dipolar evolution functions and the corresponding interprotomer distances obtained by Tikhonov regularization were determined in the presence of internal ADP (rosy brown) and in the presence of internal ATP and external K$^+$ (blue), respectively.

The following figure supplements are available for figure 6:

**Figure supplement 1.** Structural gating features of the KtrB subunit.

**Figure supplement 2.** Comparison of water accessibility in the cavity below the gating region from MD simulations of ADP- (A) and ATP-bound (B) states of KtrAB.

**Figure supplement 3.** Pulsed EPR data of spin-labeled variants KtrAB/KtrB T318CR1.

**Figure supplement 4.** Fluctuations of the intramembrane loop in MD simulations of *B. subtilis* KtrB in its ATP-bound conformation.

---

increases (*Ohwada and Sagisaka, 1987*; *Ohwada et al., 1994*), which leads to the replacement of ADP by ATP in a competitive manner. ATP binding then induces the conformational change of the octameric KtrA ring (*Albright et al., 2006*) resulting in the formation of helix hairpins of D1M2 at the membrane surface and partial unfolding of D4M2. The protein is in its activated state; hence, the gating region is unlocked, enabling conformational changes of the intramembrane loop within the pore, which in principle allows K$^+$ ions to translocate (*Figure 7*). However, instead of stochastic opening events which are enabled by the unlocking of the gating region, patch-clamp measurements on TrkAH (*Cao et al., 2013*) and whole-cell K$^+$ uptake studies on KtrAB (*Tholema et al., 1999*) suggest prolonged opening probabilities in the presence of ATP, such that additional factors are required to stabilize the active, open state. Lipids, for instance, may play an essential role in the regulatory process and their role should be further investigated. Furthermore, Na$^+$ ions (*Tholema et al., 1999*) and protons (*Stumpe et al., 1996*) have been previously described to activate the KtrAB and TrkAH systems, respectively, but their molecular role is not yet understood. In order to validate and detail the model, further research and high-resolution structures are needed. A high-resolution structure of the activated, open state would unravel the precise interactions that stabilize the open gate, and provide insights into the interplay between the permeant ion and the gate. Similarly, a high-resolution structure of the ADP-bound, inactive state would elucidate direct interactions between the KtrA subunits and the extended helices, as well as between the extended helices and the gating region. On a functional level it remains puzzling how the ratio of ATP to ADP determines the protein's activity under physiological conditions since in general an excess of ATP over ADP is assumed (*Corrigan and Gründling, 2013*; *Kim et al., 2015*). The numbers given in the literature, however, do not discriminate concentrations of free versus bound molecules, which could change the ratio significantly toward ADP. Additionally, recent data suggest another level of regulation by the binding of cyclic nucleotides to the C lobes of KtrA, which likely plays an additional role for the inactivation of the system (*Kim et al., 2015*). Yet, the resetting mechanism of KtrAB into its closed, inactivated state remains to be identified. One possible explanation is that the surrounding membrane provides

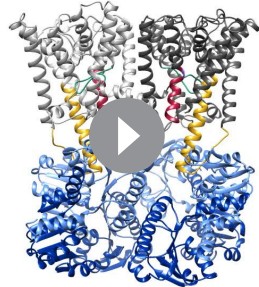

**Video 1.** Conformational differences between the ADP- and ATP-bound KtrAB complex. Linear morph between the ADP- and ATP-bound KtrAB models providing a visual aid for the undergone conformational changes shown in *Figure 6*. Subunits and structural features are colored according to *Figure 6*.

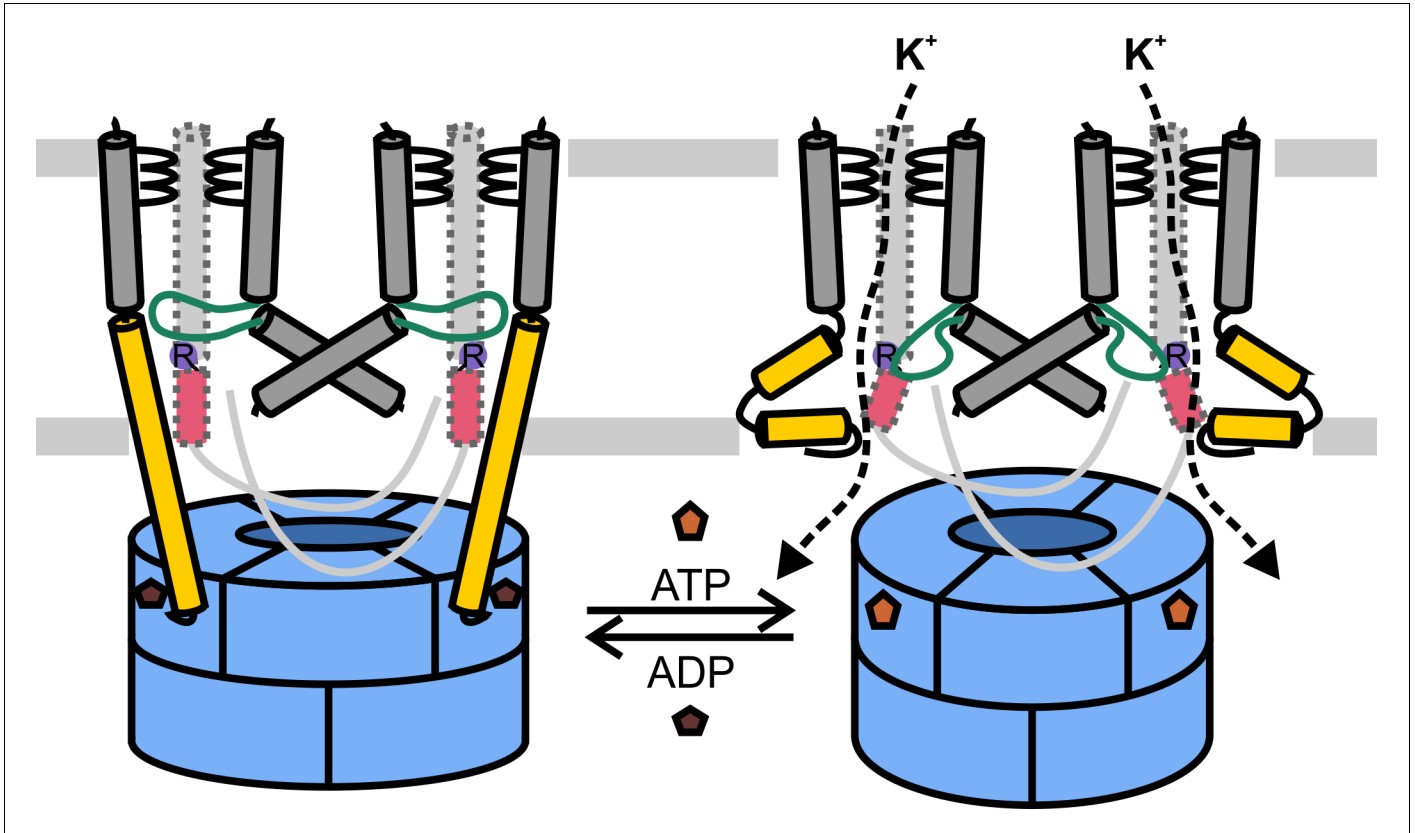

**Figure 7.** Model for KtrAB activation. Cartoons showing the ADP- and ATP-bound conformations. With ADP bound, the extended helices D1M2b of KtrB dimers interact with residues in KtrAs close to the ADP binding sites, and with the gating regions of KtrBs. The gates are locked in the closed conformation. In the presence of ATP, the D1M2 helices break, forming helix hairpins at the membrane surface, and helices D4M2b move away from the pore. The disruption of interactions in the gating regions allows the intramembrane loops to move. In general, K$^+$ flux is enabled. KtrAs, blue; KtrBs, grey with helices D1M2b, yellow; helices D4M2b, magenta; intramembrane loop, green, and conserved arginine, purple; ADP, dark brown; ATP, light brown.

the activation energy needed for this conformational change. This hypothesis seems plausible knowing that the lipid composition is adjusted upon a hyperosmotic shock (*Romantsov et al., 2009*). The deactivation of the KtrAB system might be coordinated with the activation of compatible solute transporters, which were shown to be activated by the altered lipid composition (*Culham et al., 2003*; *Peter et al., 1996*; *Rübenhagen et al., 2000*; *Schiller et al., 2006*; *van der Heide et al., 2001*). Future structural and functional studies are needed to shed light on this deactivating mechanism.

A three-stage translocation mechanism has been suggested previously for a variety of different single-pore RCK channels (*Hite et al., 2015*; *Jiang et al., 2002b*; *Kong et al., 2012*) (*Figure 8*). This translocation mechanism implies that ligand-induced conformational changes in the RCK domain are transferred via a flexible linker to the helical bundle of the pore domain. Our data suggest an unexpected, new mechanism of regulation for the Ktr/Trk family. The unique features of this family, namely the existence of two parallel pores and the non-covalent association of cytoplasmic RCK subunits require a modified regulation mechanism. The functions of the flexible linker and the helical bundle are transferred to the movement of the elongated helices and their interactions with the gating regions. This new understanding of the general mechanism of regulation provides the starting point for further investigations into the functional role of the two parallel pores in comparison to single-pore channels.

Ktr/Trk systems are key players for the homeostasis of bacterial cell physiology, encasing osmotic resistance, cellular fitness during host infection (*Alkhuder et al., 2010*; *Stingl et al., 2007*) and

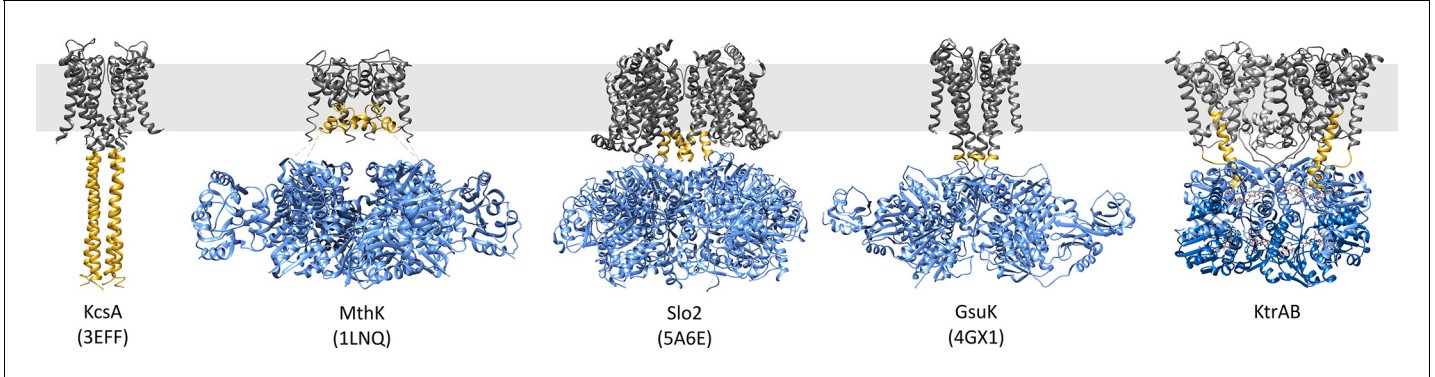

**Figure 8.** Comparison of proteins regulated by RCK domains. Side views of KcsA (PDB code 3EFF [*Uysal et al., 2009*]), MthK (PDB code 1LNQ [*Jiang et al., 2002b*]), Slo2 (PDB code 5A6E [*Hite et al., 2015*]), GsuK (PDB code 4GX1 [*Kong et al., 2012*]) and ADP-bound KtrAB. Membrane domains/protomers, grey; flexible linker or extended helices, yellow; octameric RCK domains/protomers, blue. The grey bar represents the membrane.

antimicrobial resistance (*Diskowski et al., 2015*; *Gries et al., 2013*; *Su et al., 2009*). Ktr/Trk systems are essential for bacteria but not present in mammalian cells. Therefore, the newly discovered inactive conformation may be a suitable target for structure-guided drug development of an entirely new line of antibiotics.

## Materials and methods

### Cloning and purification of KtrB and the KtrAB complex

The *ktrA* and *ktrB* genes were amplified from *Vibrio alginolyticus* genomic DNA and cloned into the pBAD18 plasmid (*Guzman et al., 1995*). $His_{10}$-ktrAB, ktrB-$His_6$ and derivates were overexpressed with 0.02% L-arabinose in *Escherichia coli* LB2003 (*Stumpe and Bakker, 1997*) in K3 minimal media (*Epstein and Kim, 1971*; *Stumpe and Bakker, 1997*; *Tholema et al., 2005*) (with 100 mg/l ampicillin and 0.2% glycerol) aerobically at 37°C for 7 hr.

For protein purification, cells were lysed by sonication in Buffer A (140 mM NaCl, 60 mM KCl, 20 mM Tris-HCl pH 8.0), the membranes were solubilized with 1% n-dodecyl-ß-D-maltoside (DDM) (>99% highly purified, GLYCON Biochemicals GmbH, Germany) for 1 hr at 4°C. Spin-cleared lysate was loaded onto a Ni-NTA Agarose affinity resin (Qiagen, Germany) supplemented with 10 mM imidazole. Nonspecifically bound proteins were removed with Buffer A containing 1.5 mM Cymal-6 (Anatrace, Maumee, Ohio) and 50 mM imidazole. The KtrAB complex was eluted with Buffer A containing 1.5 mM Cymal-6 and 500 mM imidazole and subjected to size-exclusion chromatography with a Superdex 200 Increase 10/300 GL column (GE Healthcare Life Sciences, Chicago, Illinois) preequilibrated with Buffer A containing 1.5 mM Cymal-6. For cryo-EM, the KtrAB complex was introduced via Desalting Columns (7K MWCO Thermo Fisher Scientific, Waltham, Massachusetts) to Buffer B (70 mM NaCl, 30 mM KCl, 20 mM Tris-HCl pH 8.0, 1.5 mM Cymal-6, 600 µM ADP) and finally concentrated via Centrifugal Filters (Amicon Ultra 0.5 ml MWCO 100 Merck Millipore Ltd, Billerica, Massachusetts) to 5 mg/ml.

### Isothermal titration calorimetry

Binding affinity measurements on the detergent-solubilized protein were performed on a MicroCal 200 ITC (MicroCal. Malvern, United Kingdom). 150 µl of the protein sample in Buffer A with a concentration of 8 µM were introduced into the sample cell. For the titration, the same Buffer A containing 275 µM ATP or 600 µM ADP was loaded into the injection syringe. The system was equilibrated to 25°C with a stirring speed of 750 rpm. Titration was initiated by 0.5 µl-injection, followed by 2 µl-injections every 200–250 s. In total 17 single injections were performed. Background corrections were performed by the titration of the titration solution to Buffer A, using the same experimental setup. The evaluation was performed using ORIGIN7, excluding the first injection peak.

## Cryo-EM data collection and image processing

Three microliter of His$_{10}$-KtrAB sample at a concentration of 5 mg/ml in the presence of 600 μM ADP was applied to freshly glow-discharged C-flat multi-hole carbon grids (CF-MH-4C multi C-Flat, ProtoChips, Morrisville, North Carolina) and vitrified using a Vitrobot (FEI, Hillsboro, Oregon) at 70% humidity with 10 s blotting time at 10°C. Cryo-EM images were recorded on an in-column energy-filtered JEOL 3200 FSC electron microscope with a Gatan K2 direct detector. Images were collected at a nominal magnification of 20,000x, corresponding to a calibrated pixel size of 1.63 Å, at 1.5–2.5 μm defocus in movie mode. Dose-fractionated 8 s movies of 40 frames with a total electron dose of 27 electrons per Å$^2$ were recorded. Global beam-induced motion was corrected by movie frame processing (Li et al., 2013). 35,400 particles were selected from 800 micrographs. A model calculated from the X-ray structure of ATP-bound KtrAB (PDB code 4J7C [Vieira-Pires et al., 2013]) low-pass filtered to 60 Å was used as initial model for refinement. 3D map refinement and 2D and 3D classification was carried out with RELION 1.3 (Scheres, 2012). The final model from 20,500 particles had a resolution of 6.6 Å.

## LILBID-MS

Three microliter of a detergent solubilized KtrAB sample at a concentration of 10 μM in Buffer D (25 mM NaCl, 15 mM KCl, 20 mM Tris-HCl pH 8.0 and 1.5 mM Cymal-6) was used for mass spectrometry (MS). The measurements were performed on a homebuilt reflectron-time-of-flight (TOF) setup as published in detail (Morgner et al., 2007). MS spectra show averaged signals between 500 to 1000 droplets.

## Spin labeling of KtrB and the complex KtrAB

For site-directed spin-labeling, the spin-cleared lysate was bound to the Ni-NTA Agarose affinity resin in the presence of 5 mM $\beta$-mercaptoethanol and 10 mM imidazole. The first washing step was carried out with degassed Buffer A containing 50 mM imidazole and 1.5 mM Cymal-6 for variant KtrAB A122C, or 0.04% DDM for variant KtrB T318C and variant KtrAB T318C. Protein was labeled overnight at 4°C with 1 mM (1-oxyl-2,2,5,5- tetramethylpyrrolidin-3-yl)methylmethanethiosulfonate spin label (MTSSL) (Toronto Research Chemicals, Inc., Canada) in Buffer A containing the respective detergents. To remove free spin label, the Ni-NTA Agarose affinity resin was washed with degassed Buffer A containing the respective detergent. The protein was eluted with Buffer A containing 500 mM imidazole and detergent. Subsequently, the protein was further purified by size-exclusion chromatography in combination with a pre-equilibrated Superdex 200 Increase 10/300 GL column (GE Healthcare Life Sciences) in Buffer A with the respective detergent. For the EPR measurements, the protein was either subjected to reconstitution into liposomes or for the detergent-solubilized samples concentrated via Centrifugal Filters (Amicon Ultra 0.5 ml MWCO 100 Merck Millipore Ltd) to 10 mg/ml and supplemented with 14% deuterated glycerol (v/v) (Sigma, St. Louis, Missouri) and indicated ATP and ADP concentrations, respectively.

## Reconstitution into liposomes

Spin-labeled proteins were reconstituted into liposomes containing E. coli polar lipids (prepared from Avanti total lipid extract) and egg yolk L-α-phosphatidylcholine (Sigma) in a 1:10 protein to lipid ratio (w/w). Liposomes were prepared as described before in a 3 + 1 (w/w) ratio (Hänelt et al., 2010b) containing 1 mM ATP and ADP, respectively, as indicated. The liposomes were diluted with Buffer C (200 mM Choline-Cl, 20 mM Tris-HCl pH 8.0) to 4 mg/ml and titrated with Triton X-100 (Sigma). The detergent-destabilized liposomes were mixed with spin-labeled protein and incubated for 30 min at room temperature under gentle agitation. Polystyrene beads (Biobeads SM2) were added at a wet weight of 40 mg/ml, and the sample was incubated for further 15 min at room temperature. Subsequently, fresh Biobeads SM2 (40 mg/ml) were added four times with incubations at 4°C of 15 min, 30 min, overnight, and 1 hr, respectively. The beads were removed, and the mixture was diluted threefold with Buffer C. After collecting the proteoliposomes by ultracentrifugation, they were washed twice with buffer C. Finally, the proteoliposomes were dissolved in buffer C and supplemented with 14% deuterated glycerol (v/v) (Sigma) and when indicated with 100 mM KCl for further pulsed EPR experiments.

## Pulsed EPR spectroscopy

For pulsed EPR measurements recorded at X band (9.4 GHz), 30–40 µl of the sample was loaded into EPR quartz tubes with a 3 mm outer diameter and shock frozen in liquid nitrogen. The experiments were performed at 50 K on an Elexsys 580 spectrometer (Bruker, Billerica, Massachusetts). Temperature was controlled by the utilization of a continuous-flow helium cryostat (Oxford Instruments, United Kingdoms) and a temperature controller (Oxford Instruments). The four-pulse DEER sequence was applied (*Pannier et al., 2011*). The previously established parameters (*Hänelt et al., 2010b*) were kept unmodified with observer pulses of 16–32 ns and a pump pulse of 12 ns. The frequency of the pump pulse was set to the maximum of the nitroxide EPR spectrum and the frequency separation to 65 MHz. For pulsed EPR measurements recorded at 50K and at Q band (34 GHz), 15 µl of the sample was placed into EPR quartz tubes with a 1.6 mm outer diameter and shock frozen in liquid nitrogen. The four-pulse DEER sequence was applied with observer pulses of 32 ns and a pump pulse of 20 ns. The frequency separation was set to 50 MHz and the frequency of the pump pulse to the maximum of the nitroxide EPR spectrum. Validation of the distance distributions were performed by means of the validation tool included in DeerAnalysis (*Jeschke et al., 2006*). Here, the parameters Background start and Background density were varied in the suggested range by applying fine grid resulting in 121 trials. Afterwards, poor fits were excluded by using a prune level of 1.15. Moreover, interspin distance prediction was performed using the rotamer library approach included in the MMM software package (*Polyhach et al., 2011*). The calculation of the interspin distance distributions is based on the *V. alginolyticus* ADP- and ATP-bound $KtrB_2A_8B_2$ models generated in this study for the comparison with the experimentally determined interspin distance distribution of the ATP- and ADP-bound conformation, respectively.

## Homology modeling

A set of sequence homologs of KtrAB from *V. alginolyticus* was obtained by using NCBI BLAST against the non-redundant sequence database. The sequences were clustered using the CD-HIT (*Huang et al., 2010*) webserver based on the sequence similarity, in which sequences with more than 70% of sequence identity were put in one cluster. In each cluster, a representative sequence was kept and other sequences were discarded. This procedure resulted in 116 sequences. Multiple sequence alignments of KtrA and KtrB homologs were then generated using PSI/TM-Coffee (*Chang et al., 2012*). The initial multiple sequence alignments were accordingly refined by (i) considering the pairwise alignments between KtrA/B sequences from *V. alginolyticus* and *B. subtilis*, generated using AlignMe (*Stamm et al., 2014*) and (ii) removing gaps in the secondary structure elements and the transmembrane regions. The KtrAB system from *V. alginolyticus* was modeled based on the ATP-bound X-ray crystal structure of KtrAB system from *B. subtilis* (PBD code 4J7C [*Vieira-Pires et al., 2013*]) using Modeller 9v16 premised on the final sequence alignment (*Sali and Blundell, 1993*). Among 100 candidates, the model with the highest Modeller score was selected for further analysis.

## Molecular dynamics simulations

We performed all-atom explicit solvent MD simulations of lipid membrane-embedded KtrAB, starting from the homology model. According to MCCE (*Alexov and Gunner, 1997*) electrostatics calculations for neutral pH, all Asp, Glu, Arg and Lys residues were charged. All His residues were neutral, protonated at either their Nδ or Nε atoms according to the network of hydrogen bonds. The poorly resolved C-lobe part of KtrA was not included in the model.

ATP-bound KtrAB was embedded (*Wu et al., 2014*) in a bilayer of 554 palmitoyl-oleoyl-phosphatidylcholine (POPC) lipids. The protein was hydrated with 150 mM KCl electrolyte, resulting in a box size of ~15 × 15 × 14 $nm^3$ containing 300,024 atoms. The all-atom CHARMM36 force field was used for the protein, lipids and ions, and TIP3P was used for water molecules (*Best et al., 2012*; *Jorgensen et al., 1983*; *Klauda et al., 2010*). The MD trajectories were analyzed with Visual Molecular Dynamics (VMD) (*Humphrey et al., 1996*).

To improve the statistics and assess the consistency, for each system described in the paper two independent sets of simulations were performed. In these simulations, two different software packages were used, NAMD and GROMACS, because they offer different functionalities and operate at different computing speeds. Importantly, the results obtained with the two simulation packages are

fully consistent. For a better overview, the MD simulation setups are summarized in *Table 1*. In the analysis of water accessibility, we report the combined results of both setups (NAMD and GROMACS) for each system.

A first set of MD simulations of the two systems was performed using NAMD 2.9 (*Phillips et al., 2005*). After 10,000 steps of conjugate gradients energy minimization, 10 ns of MD simulation were carried out, in which all non-hydrogen atoms of the protein were constrained to their initial positions using springs with progressively smaller force constants, starting at 15 kcal·mol$^{-1}$·Å$^2$. The analysis was carried out on unconstrained simulations. Periodic boundary conditions were used, with particle-mesh Ewald electrostatics, a 1.2 nm non-bonded cutoff, a time step of 2 fs, SHAKE constraints on all bond lengths (*Ryckaert et al., 1977*), and a constant temperature of 310 K maintained by a Langevin thermostat (*Adelman and Doll, 1976*) with a coupling coefficient of 1.0 ps$^{-1}$. A Nosé–Hoover Langevin barostat (*Feller et al., 1995*) was used to apply a constant pressure of 1 bar normal to the membrane plane. The ratio of the membrane in the x-y plane was kept constant while allowing fluctuations along all axes.

A second set of simulations was performed using GROMACS 5.0 (*Abraham et al., 2015*). The starting systems were minimized for 50,000 steps with steepest descent energy minimization and equilibrated for 10 ns of MD simulation in the NPT ensemble in which all non-hydrogen atoms of the protein were restrained to the fixed reference positions with progressively smaller force constants, starting at 1000 kJ·mol$^{-1}$·nm$^2$. Analysis was carried out on unconstrained simulations. Periodic boundary conditions were used. Particle mesh Ewald (*Darden et al., 1993*) with cubic interpolation and a 0.16 nm grid spacing for Fast Fourier Transform was used to treat long-range electrostatic interactions. The time step was two fs. The LINCS algorithm (*Hess et al., 1997*) was used to fix all bond lengths. Constant temperature (310 K) was set with a Nosé-Hoover thermostat (*Hoover, 1985*), with a coupling constant of 0.5 ps. A semiisotropic Parrinello-Rahman barostat (*Parrinello and Rahman, 1981*) was used to maintain a pressure of 1 bar.

All-atom explicit solvent molecular dynamics (MD) simulations were also carried out for the lipid-membrane embedded KtrB of *B. subtilis*, starting from the ATP-bound X-ray structure (PBD code 4J7C [*Vieira-Pires et al., 2013*]). The procedure is the same as described for the homology models. Briefly, KtrB was embedded in a bilayer of 509 POPC lipids and hydrated with 150 mM KCl electrolyte, resulting in a box of ~14.6 × 14.6 × 10 nm$^3$ (195,255 atoms) in size. The simulations were performed using GROMACS 5.0 and NAMD 2.9 with a length of 400 ns and 240 ns, respectively. Simulations of ADP-bound KtrAB are described at the end of the following section.

## Molecular dynamics flexible fitting (MDFF) simulations

In the MDFF simulations, the homology model of KtrB$_2$A$_8$B$_2$ was used. The MDFF plugin v0.4 in VMD1.9.2 was used to perform molecular dynamics simulations guided by the cryo-EM density map (*Trabuco et al., 2008*). The MDFF simulations were performed in vacuum and with a g-scale factor of 0.3, which describes the strength of the external potential derived from the EM density map. Initially, the KtrB$_2$A$_8$B$_2$ model was docked rigidly into the cryo-EM density with Situs 2.7.2 (*Wriggers, 2012*) (colores module). Afterwards, a 500-ps MDFF simulation was performed followed by

**Table 1.** Description of the different MD simulation systems and setups performed in this study.

| Simulation code | System | Conformation | Organism | MD package | Length |
| --- | --- | --- | --- | --- | --- |
| ABTG$_{VA}$ | KtrAB | ATP-bound | *V. alginolyticus* | GROMACS | 530 ns |
| ABTN$_{VA}$ | KtrAB | ATP-bound | *V. alginolyticus* | NAMD | 200 ns |
| ABDG$_{VA}$ | KtrAB | ADP-bound | *V. alginolyticus* | GROMACS | 450 ns |
| ABDN$_{VA}$ | KtrAB | ADP-bound | *V. alginolyticus* | NAMD | 200 ns |
| BTG$_{BS}$ | KtrB | ATP-bound | *B. subtilis* | GROMACS | 400 ns |
| BTN$_{BS}$ | KtrB | ATP-bound | *B. subtilis* | NAMD | 240 ns |
| ABTGB$_{VA}$ | KtrAB | ATP-bound | *V. alginolyticus* | GROMACS | 10 × 60 ns |
| ABDGB$_{VA}$ | KtrAB | ADP-bound | *V. alginolyticus* | GROMACS | 10 × 60 ns |

5000 steps of energy minimization. Secondary structure, chirality, and *cis*-peptide restraints were used during the MDFF simulation in order to enforce the original secondary structure, chirality, and isomerism. Additional domain restraints were used for the D2M1 helix during the MDFF simulation to maintain its integrity. The density cross-correlation coefficient, used as fit quality indicator, improved from 0.70 in the initial model to 0.89. The D1M2 and D4M2 helices could not be directly fitted from the broken form of the ATP-bound conformation into the continuous helix of the ADP-bound state. Therefore, these helices were modeled as a continuous helix using Modeler (*Sali and Blundell, 1993*) and then this refined model was used for fitting.

Afterwards, all-atom simulation systems were set up for the EM-fitted model similar to the conditions described above for the ATP-bound model. Briefly, KtrAB was embedded in a bilayer of 555 POPC lipids, and hydrated with 150 mM KCl electrolyte. MD simulations in NAMD 2.9 and GROMACS 5.0 were 200 and 450 ns long, respectively.

### Blind loop modeling

To assess the role of the intramembrane loop on the hydration state of the gating region, additional starting structures were created in which this loop was modeled blindly. In this way, the KtrAB system apart from the loop (residue 315 to 323) from *V. alginolyticus* was modeled based on the ATP-bound X-ray crystal structure of the KtrAB system from *B. subtilis* (PBD code 4J7C [*Vieira-Pires et al., 2013*]), whereas the loop was modeled de novo by using Modeller 9v16. Ten models were selected in which the mean distance for T318 probes was close to the experimental value of the pulsed EPR measurements. Following the protocol described above, 60-ns long MD simulations using GROMACS were performed for each of these models.

### Growth complementation assay

The growth complementation assays were performed as previously described (*Tholema et al., 2005*).

## Acknowledgements

IH thanks Prof. Evert Bakker for his mentorship. We thank Prof. Werner Kühlbrandt for his support of the cryo-EM and for his comments on the manuscript. Prof. Thomas Prisner and Dr. Burkhard Endeward are acknowledged for their support on the EPR measurements. We thank Edoardo D'Imprima for discussions and Drs. Juan Castillo-Hernández and Özkan Yildiz for computer support. This work was supported by the Max Planck Society and by the German Research Foundation via Emmy Noether grant HA 6322/3–1 (I.H.), HA 6322/2–1 (IH), SFB 807 Membrane Transport and Communication (ARM, NM, GH and IH), and the Cluster of Excellence Frankfurt (Macromolecular Complexes) (ARM and IH). The cryo-EM map was deposited in the Electron Microscopy Data Bank with accession code EMD-3450.

## Additional information

### Funding

| Funder | Grant reference number | Author |
|---|---|---|
| Deutsche Forschungsge-meinschaft | HA 6322/3-1 | Inga Hänelt |
| Max-Planck-Gesellschaft | | Ahmad Reza Mehdipour<br>Deryck J Mills<br>Gerhard Hummer<br>Janet Vonck |
| Deutsche Forschungsge-meinschaft | HA 6322/2-1 | Inga Hänelt |
| Deutsche Forschungsge-meinschaft | SFB 807 | Nina Morgner<br>Gerhard Hummer<br>Inga Hänelt |
| Deutsche Forschungsge- | CEF Macromolecular | Ahmad Reza Mehdipour |

| meinschaft | Complexes | Inga Hänelt |

The funders had no role in study design, data collection and interpretation, or the decision to submit the work for publication.

### Author contributions
MD, ARM, Formal analysis, Validation, Investigation, Visualization, Methodology, Writing—original draft; DW, Formal analysis, Validation, Investigation, Visualization, Methodology, Writing—review and editing; DJM, Investigation, Methodology; VM, Investigation, Writing—review and editing; NB, Validation, Investigation, Writing—review and editing; JH, Formal analysis, Investigation, Methodology; NM, H-JS, Formal analysis, Validation, Methodology, Writing—review and editing; GH, Supervision, Funding acquisition, Validation, Investigation, Methodology, Writing—original draft; JV, Formal analysis, Supervision, Validation, Investigation, Visualization, Methodology, Writing—original draft; IH, Conceptualization, Formal analysis, Supervision, Funding acquisition, Validation, Visualization, Methodology, Writing—original draft, Project administration

### Author ORCIDs
Nina Morgner, http://orcid.org/0000-0002-1872-490X
Janet Vonck, http://orcid.org/0000-0001-5659-8863
Inga Hänelt, http://orcid.org/0000-0003-1495-3163

## Additional files

### Major datasets
The following dataset was generated:

| Author(s) | Year | Dataset title | Dataset URL | Database, license, and accessibility information |
| --- | --- | --- | --- | --- |
| Marina Diskowski, Deryck J Mills, Natalie Bärland, Inga Hänelt, Janet Vonck | 2016 | Structure of the K+ transporter KtrAB from Vibrio alginolyticus in the ADP-bound state | http://www.ebi.ac.uk/pdbe/entry/emdb/EMD-3450 | Publicly available at EBI Protein Data Bank (accession no: EMD-3450) |

The following previously published dataset was used:

| Author(s) | Year | Dataset title | Dataset URL | Database, license, and accessibility information |
| --- | --- | --- | --- | --- |
| Vieira-Pires RS, Morais-Cabral JH | 2013 | KtrAB potassium transporter from Bacillus subtilis | http://www.rcsb.org/pdb/explore/explore.do?structureId=4J7C | Publicly available at Protein Data Bank (accession no: 4J7C) |

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
