## [Decision Letter]

Thank you for submitting your article "Helical jackknives control the gates of the double-pore K^+^ transporter KtrAB" for consideration by *eLife*. Your article has been reviewed by three peer reviewers, and the evaluation has been overseen by Kenton Swartz as the Reviewing Editor and Richard Aldrich as the Senior Editor. The reviewers have opted to remain anonymous.

The reviewers have discussed the reviews with one another and the Reviewing Editor has drafted this decision to help you prepare a revised submission.

Summary:

Diskowski et al. convincingly show that the mechanism of inactivation of the KtrAB K^+^ channel differs from those of other K^+^ channels regulated by RCK domains. Specifically, the authors report an ADP-bound structure of the KtrAB complex from *V. alginolyticus*, obtained through cryo-EM at 6-7 A resolution, and assumed to reflect the inactive state of the channel. When compared with the ATP-bound structure, reported elsewhere, this structure indicates that inactivation results from a series of structural changes in the RCK domains (upon displacement of ATP by ADP), which in turn foster/permit changes in the secondary – and tertiary structures of the channel domain, so as to close off the permeation pathway on the intracellular side. The authors speculate that this mechanism differs from those of other RCK-regulated channels because the RCK domains are not directly linked to the channel domain, which is plausible though not self-evident. In any case, the mechanism inferred from the structure is undoubtedly novel and would be of interest to the membrane biophysics community. Importantly, key aspects of the proposed mechanism are also supported, at least qualitatively, by complementary EPR spectroscopy measurements. Specifically, the EPR data indicates that a flexible loop known to serve as a reversible intracellular gate in the ATP-bound, active state of the channel becomes more immobilized in the ADP-bound state – consistent with the notion that inactivation results from a more robust structural occlusion of the permeation pathway. EPR measurements are also qualitatively consistent with the proposal that one of the helices in the channel domain has the capacity to adopt multiple configurations, and that ADP binding favors the more extended form, as seen in the cryo-EM structure.

Essential revisions:

1) The reviewers concur that your claim that "the ATP-bound conformation is most likely the preferred state of KtrB in the absence of the RCK domain", i.e. that "in the absence of the [Ktr]A subunits, the extended α-helix of the ADP-bound conformation" is unstable, is not directly supported by the simulation data currently provided in the manuscript (Figure 5/Figure 5—figure supplement 1). Therefore, the reviewers request that you either provide a calculation specifically designed to probe the relative stability of these two conformations (i.e. a free-energy difference calculation, or a sufficiently long unbiased simulation where the two conformations are sampled spontaneously), or that you remove these statements and the related simulation data. Similarly, the reviewers question your conclusions regarding the dynamics of the gating loop, and its impact on the degree of hydration of the channel interior (Figure 6 and Figure 6—figure supplement 2), given that the structure of this loop is not resolved in the proposed inactive state – implying the loop was modeled, somehow. Thus, in order for the reviewers to be able to evaluate your conclusions, they request that you explain how this loop was modeled, and that you provide evidence of the validity or expected accuracy of your approach – for example by blind-modeling the structure of this loop in active KtrB, and comparing the prediction with the actual experimental result. Alternatively, you might opt to also remove this portion of the study. If you decide to preserve the simulation component, the reviewers also request that you describe the similarities and differences between the two simulations conducted for each molecular system, and that you clarify what the source of the data is for each of your figures.

2) In the subsection “Helix D1M2 connects KtrB with KtrA” and Figure 5, the Q band DEER data (the dipolar evolution traces) have very poorly defined features which reflect rather fast T1 relaxations. An issue that is particularly worrisome on the mutant at position 122. This is reflected in the calculated distance distributions, which give rise to broad and overlapping peaks and it might be a consequence of conformational heterogeneity at position 122. In contrast, however subtle, the differences in raw data for the dipolar evolution at position 318 that monitor gate opening, are more believable and reflect tighter distance distributions (in spite of the increased noise). Also, surprisingly, the X bad DEER data at position 318 display easily interpretable features in spite of the increased noise. We request that the authors consider alternatives to the results at position 122. We would recommend considering different labeling site at helix D1M2 with improved dipolar evolution behavior to bring this home. Although the existing DEER data does not go against the overall gating model, given a single credible set of DEER distance distributions it would be a stretch to claim this data as a direct demonstration of the present model. The authors should also openly discuss the problems with their data at position 122 and temper their conclusions accordingly.

3) The model does not actually explain how the channel is activated. To their credit the authors acknowledge this shortcoming, and suggest that activation may be due to a range of additional, unidentified, interactions, including a role of the lipid membrane, which they say changes in response to osmotic shock. We were left wondering whether they tried to measure their EPR distances on helices D1M2 in liposomes of different lipid composition? They do for the gating region, (Figure 5). In the case of BetP, which is also regulated in response to osmotic shock, specific anionic lipids play key roles in regulating structure transitions within the transporter. We wonder if something similar is happening here? Have the authors tried to determine the structure in the presence of specific *Vibrio alginolyticus* lipids? These may have been stripped during purification?

4) The authors should also clarify what is known about the gating properties of the channel. Once activated, will it remain in an open state for long periods of time? If so, one would expect some structural stabilization of the gate region. The absence of such an interaction from the crystal structures and EM structure here suggests some component of the system may be missing in these studies, possibly lipid. Currently it reads to me as though the binding of the ATP puts the channel in a state that is able to stochastically sample the open state in response to electrostatic interactions with K ions – yet the paper implies that ADP-ATP exchange causes structural rearrangements that lock, or stabilize the closed/open states via the linking helices. The authors should better develop what is known about the gating properties of the channel and discuss their model in that context, clearly pointing out ambiguities that remain.

[Editors' note: further revisions were requested prior to acceptance, as described below.]

Thank you for resubmitting your work entitled "Helical jackknives control the gates of the double-pore K^+^ uptake system KtrAB" for further consideration at *eLife*. Your revised article has been favorably evaluated by Richard Aldrich (Senior Editor), Kenton Swartz (Reviewing Editor), and two of the original reviewers.

The Reviewing editor and reviewers agree that the revised manuscript is improved and that the work is appropriate for *eLife*. However, reviewer #1 feels strongly that the data in Figure 5/Figure 5—figure supplement 1 and the statements in the second paragraph of the subsection “Helix D1M2 connects KtrB with KtrA” should be removed because they are not valid, as pointed out in the original critiques. His/her request is elaborated below. We would be pleased to receive a suitably revised manuscript and proceed with acceptance of the manuscript.

*Reviewer #1:*

The authors have satisfactorily addressed the issues and questions I raised in my original review, except point #1. The authors reply that:

"[…] we agree that the thermodynamic stability of the different D1M2 conformations cannot be inferred quantitatively from our MD simulation data. Full-fledged free energy simulations and/or sampling of reversible transitions of such large-scale changes are not feasible. We therefore followed the recommendation and toned down the discussion."

I disagree with the authors' view that it is unfeasible to carry out a free-energy simulation to probe the proposed conformational change in the D1M2 helix, in the absence of KrtA, to demonstrate whether the extended conformation requires KtrA to be stable, or whether this region favors instead the "broken" form. However, I recognize now as I did originally that this isn't a trivial task, and that it might be beyond the expected scope of this study. It is equally clear to me, though, that the existing simulation data (Figure 5/Figure 5—figure supplement 1) does not provide any meaningful information on this matter. There's no direct relationship between the relative stability of these two states and the descriptor calculated from the simulation (the RMSF for either state) – neither quantitatively or qualitatively. It is for these reasons that our original request was that either a suitable calculation is presented, or that "you remove these statements and the related simulation data", meaning the data in Figure 5/Figure 5—figure supplement 1 and the statements in the second paragraph of the subsection “Helix D1M2 connects KtrB with KtrA”. It was not simply recommended that the discussion be toned down. Therefore, I cannot recommend the publication of this manuscript in *eLife* in its current form. I would be more than willing to do so if the authors complied with this straightforward request – which is no way detrimental for this study.

*Reviewer #3:*

The main conclusions of this paper are based on the cryo EM structure and supporting information from EPR and MD. I think that overall the authors have tried to address the main points and have toned down their conclusions based on the MD analysis. Our suggestion that additional sites be tested in the EPR section proved difficult to obtain, as expected, although the authors have presented data supporting their statement that most residues in the extended helix gave poor or difficult to interpret distance distributions. The authors however stand by their claim that the data qualitatively supports their argument of a structural transition in the D1M2 region following ATP/ADP binding and I would agree with this conclusion. On my specific query regarding the role of lipids, they didn't actually address, instead performing a set of additional experiments in detergent measuring the flexibility in the presence of nucleotide, which does support their arguments, but didn't address the more important question on the role of lipids in stabilizing the D1M2 region. This was never going to be a deal breaker anyway.

I am still in favor of this paper being published, as the structural work does present novel insights into the gating mechanism within this K^+^ channel family. The 6 A maps are more than clear enough to propose the jackknife model from a structural perspective. The authors have made suitable changes to the text and data to better support their conclusions, which I think are, on the whole, justified.

---

## [Author Response]

*Essential revisions:*

*1) The reviewers concur that your claim that "the ATP-bound conformation is most likely the preferred state of KtrB in the absence of the RCK domain", i.e. that "in the absence of the [Ktr]A subunits, the extended α-helix of the ADP-bound conformation" is unstable, is not directly supported by the simulation data currently provided in the manuscript (Figure 5/Figure 5—figure supplement 1). Therefore, the reviewers request that you either provide a calculation specifically designed to probe the relative stability of these two conformations (i.e. a free-energy difference calculation, or a sufficiently long unbiased simulation where the two conformations are sampled spontaneously), or that you remove these statements and the related simulation data.*

In response, we agree that the thermodynamic stability of the different D1M2 conformations cannot be inferred quantitatively from our MD simulation data. Full-fledged free energy simulations and/or sampling of reversible transitions of such large-scale changes are not feasible. We therefore followed the recommendation and toned down the discussion. In particular, we now emphasize the higher flexibility of the D1M2 region in the absence of KtrA, which can be deduced from the enhanced RMSF values.

*Similarly, the reviewers question your conclusions regarding the dynamics of the gating loop, and its impact on the degree of hydration of the channel interior (Figure 6 and Figure 6—figure supplement 2), given that the structure of this loop is not resolved in the proposed inactive state – implying the loop was modeled, somehow. Thus, in order for the reviewers to be able to evaluate your conclusions, they request that you explain how this loop was modeled, and that you provide evidence of the validity or expected accuracy of your approach – for example by blind-modeling the structure of this loop in active KtrB, and comparing the prediction with the actual experimental result. Alternatively, you might opt to also remove this portion of the study.*

In response, we now provide evidence that the detailed conformation of the intramembrane loop has a negligible effect on the hydration level of the gating region. Water would enter into the gating region from the cytoplasmic side and this entry would be mainly controlled by the conformation of D1M2 and D4M2.

Originally, the intramembrane loop was modeled like the rest of the structure based on the X-ray structure from *B. subtilis.* Although we did not include any additional experimental data for the modelling, both ATP- and ADP-bound models have a good agreement with the EPR data for the T318 residue, as can be seen in Figure 9.

Author response image 1.**DOI:**
http://dx.doi.org/10.7554/eLife.24303.020

Following the suggestion of Editor and reviewers, we now check the effect of possible changes in the conformation of loop on the hydration level by generating additional blindly modeled loops. Whereas the rest of the structure was modeled based on the X-ray structure from *B. subtilis,* the loop region (residues 315 to 323) was modeled de novo using the program Modeller. Ten models each in ADP- and ATP-bound states were selected according to their consistency with T318 EPR data. We then performed simulations for each of these models. The results are included in the manuscript (subsection “Nucleotide-dependent K^+^ 183 gating”, last paragraph, Figure 6) and show that the loop conformation has a minimal effect on the hydration level.

*If you decide to preserve the simulation component, the reviewers also request that you describe the similarities and differences between the two simulations conducted for each molecular system, and that you clarify what the source of the data is for each of your figures.*

The different MD simulation systems and setups performed in this study are now summarized in the new Table 1 and indicated in each figure legend.

*2) In the subsection “Helix D1M2 connects KtrB with KtrA” and Figure 5, the Q band DEER data (the dipolar evolution traces) have very poorly defined features which reflect rather fast T1 relaxations. An issue that is particularly worrisome on the mutant at position 122. This is reflected in the calculated distance distributions, which give rise to broad and overlapping peaks and it might be a consequence of conformational heterogeneity at position 122. In contrast, however subtle, the differences in raw data for the dipolar evolution at position 318 that monitor gate opening, are more believable and reflect tighter distance distributions (in spite of the increased noise). Also, surprisingly, the X bad DEER data at position 318 display easily interpretable features in spite of the increased noise. We request that the authors consider alternatives to the results at position 122. We would recommend considering different labeling site at helix D1M2 with improved dipolar evolution behavior to bring this home. Although the existing DEER data does not go against the overall gating model, given a single credible set of DEER distance distributions it would be a stretch to claim this data as a direct demonstration of the present model. The authors should also openly discuss the problems with their data at position 122 and temper their conclusions accordingly.*

We agree that position 122 shows rather poor features with respect to a defined oscillation in the dipolar evolution traces compared to data from position 318. This result, however, would probably be the same for every single position in the extended helix, since our measurements are not solely restricted on the spin-spin distances of the KtrB homodimer but additionally include the interdimer distances within the KtrB_2_A_8_B_2_ sandwich. We have clarified this point in the revised manuscript (“Based on a MMM evaluation (Polyhach et al., 2011) of the different positions within our models’ D1M2 region, we selected the position that potentially undergoes the largest nucleotide-regulated conformational changes…”) and predicted distance distributions for the sum of all three possible distance distributions (Figure 5—figure supplement 3). The sum of multiple distance contributions explains the noticeable broad distance distributions and the limited oscillations in the corresponding dipolar evolution traces, both in the presence of ADP or ATP. Nonetheless, the dipolar evolution traces in the presence of ADP or ATP are significantly different from each other, clearly reflecting a conformational change from short to long distances. At this point, examination of the raw data is actually more reliable and conclusive than evaluation of the final distance distributions.

For further clarification, we have repeated our measurements at very high ADP or ATP concentrations of 100 mM, as well as in the absence of additional nucleotides (cf. Figure 5—figure supplement 4). The measurements show a broad distance distribution in the absence of nucleotides, which indicates that the extended helix adopts different intermediates ranging from the ADP- to the ATP-bound conformation. At high ADP concentrations the fast decay in the dipolar evolution trace gets significantly more pronounced. In comparison to ADP-containing measurements, the ATP-containing measurements reveal a slower decay, pointing towards significantly reduced probability of short distances. With an increase of ATP concentration in the sample this effect becomes even more severe. However, in the presence of ATP the effect is less pronounced in the experimentally determined distance distributions, as one would expect. This observation is consistent with the fact that the d2 time of 3 µs provides reliable distance distributions, and consequently reliable distance probabilities with mean interspin distances only up to approximately 5.4 nm. Thus, the probability for the long distance could be even higher and the exact distance could be even longer, which in addition has been demonstrated by the validation tool. Taken together, the new Figure 5—figure supplement 4 reveals that in the apo-state the KtrAB conformation is more likely in a dynamic equilibrium. Furthermore, the direction of the conformational changes, towards small or long distances, is dependent on the nucleotide concentration. This is exactly what one would expect for a system that is regulated by competitive ligand binding.

As proposed by the reviewers we, in fact, analyzed additional residues within the extended helix. However, as given by the MMM analysis (Figure 10), most residues even in the ADP-bound conformation are too far away from each other to be properly resolved using the d2 time of 3-4 µs. Consequently, they would be indistinguishable from the distance distribution in the presence of ATP.

Author response image 2.**DOI:**
http://dx.doi.org/10.7554/eLife.24303.021

Only three residues within the D1M2 region at positions 117, 118, and 122 were potentially suitable for further investigations. Cysteine variant 117 turned out to be instable upon purification. Spin-labeled variant 118R1 was in fact measured. However, due to restrictions in the T2 relaxation time variant Q118CR1 could only be measured with a d2 time of 2.8 µs, resulting in a reliable distance distribution range up to approximately 5.2 nm. Therefore, nucleotide-induced distance changes between the ADP- and ATP-bound conformations are difficult to resolve. In the dipolar evolution traces small changes are visible and the associated distance distributions indicate a loss of shorter distances in the presence of 1 mM ATP, however these data are less clear than those of variant 122R1. Thus, we decided not to include the data for variant Q118CR1 into the manuscript.

Author response image 3.**DOI:**
http://dx.doi.org/10.7554/eLife.24303.022

Moreover, the aim of the EPR measurements in combination with MD simulations and the cryo-EM map was to proof the general existence of both conformations in dependence of the different nucleotides rather than gaining exact distances. We believe it is worth mentioning that evaluation of our new model using molecular replacement revealed densities for the extended helices in the crystallographic data deposited for the ADP-bound KtrAB structure from *Bacillus subtilis* by Szollosi and colleagues (pdb code 5BUT) in 2016. Apparently, the extended helices have been missed in their analysis because of the structure’s low resolution (6 Å) and the poor fit of the ATP-bound structure in this area, which was used for molecular replacement. We thus hypothesize that the extended helices are highly conserved among species and represent a general feature of gating regulation.

*3) The model does not actually explain how the channel is activated. To their credit the authors acknowledge this shortcoming, and suggest that activation may be due to a range of additional, unidentified, interactions, including a role of the lipid membrane, which they say changes in response to osmotic shock. We were left wondering whether they tried to measure their EPR distances on helices D1M2 in liposomes of different lipid composition? They do for the gating region, (Figure 5). In the case of BetP, which is also regulated in response to osmotic shock, specific anionic lipids play key roles in regulating structure transitions within the transporter. We wonder if something similar is happening here? Have the authors tried to determine the structure in the presence of specific Vibrio alginolyticus lipids? These may have been stripped during purification?*

We believe that now we are one step further in understanding the role of nucleotide binding to KtrAB, which is the topic of this paper: ATP binding to KtrA unlocks the cytoplasmic gate (intramembrane loop) within KtrB, while bound ADP keeps it in a closed conformation. KtrA thus functions as a gate keeper, which is also supported by the recently described inactivating function of cyclic di-AMP binding to the C lobes of KtrA in *B. subtilis* and other Gram positive bacteria. Further stimuli, as the previously discussed lipids or Na^+^, which has been described in earlier studies, are essential for stabilization of an open conformation. The elucidation of the effect of different lipid compositions goes far beyond the scope of the present study, as it will involve the establishment of reliable functional in vitro and in vivo assays. We agree with the reviewers that this is a very exciting aspect of future research, which we will surely investigate! Instead of further addressing the impact of different lipid compositions, we have now performed measurements on spin-labeled variant KtrAB-318R1 in detergent solution to investigate the mobility of the intramembrane loop dependent on the present nucleotide (Figure 6—figure supplement 3). Those measurements, as well as those with proteoliposomes, show the increased flexibility of the intramembrane loop in the presence of ATP, which supports our hypothesis that the gate gets unlocked. In addition, we repeated our measurements of reconstituted variant KtrAB-318R1 using Q-band to gain a better signal-to-noise ratio (Figure 6). A comparison of the measurements in detergent solution and in liposomes clearly shows the same trend, but also indicates a regulatory role of the lipid environment, as the occupied distances are shifted and seemingly more defined in the presence of lipids. Nevertheless, we decided to include the data of reconstituted variant KtrAB-318R1 in support of our discussion that additional stimuli are needed for complete KtrAB activation.

*4) The authors should also clarify what is known about the gating properties of the channel. Once activated, will it remain in an open state for long periods of time? If so, one would expect some structural stabilization of the gate region. The absence of such an interaction from the crystal structures and EM structure here suggests some component of the system may be missing in these studies, possibly lipid. Currently it reads to me as though the binding of the ATP puts the channel in a state that is able to stochastically sample the open state in response to electrostatic interactions with K ions – yet the paper implies that ADP-ATP exchange causes structural rearrangements that lock, or stabilize the closed/open states via the linking helices. The authors should better develop what is known about the gating properties of the channel and discuss their model in that context, clearly pointing out ambiguities that remain.*

We have addressed this point in the Results and Discussion. Furthermore, we emphasized the role of nucleotide binding in the main text. Additionally, we summarized in more detail what is known by now and what information is missing providing further research routes.

[Editors' note: further revisions were requested prior to acceptance, as described below.]

*[…] Reviewer #1:*

*The authors have satisfactorily addressed the issues and questions I raised in my original review, except point #1. The authors reply that:*

*"[…] we agree that the thermodynamic stability of the different D1M2 conformations cannot be inferred quantitatively from our MD simulation data. Full-fledged free energy simulations and/or sampling of reversible transitions of such large-scale changes are not feasible. We therefore followed the recommendation and toned down the discussion."*

*I disagree with the authors' view that it is unfeasible to carry out a free-energy simulation to probe the proposed conformational change in the D1M2 helix, in the absence of KrtA, to demonstrate whether the extended conformation requires KtrA to be stable, or whether this region favors instead the "broken" form. However, I recognize now as I did originally that this isn't a trivial task, and that it might be beyond the expected scope of this study. It is equally clear to me, though, that the existing simulation data (Figure 5/Figure 5—figure supplement 1) does not provide any meaningful information on this matter. There's no direct relationship between the relative stability of these two states and the descriptor calculated from the simulation (the RMSF for either state) – neither quantitatively or qualitatively. It is for these reasons that our original request was that either a suitable calculation is presented, or that "you remove these statements and the related simulation data", meaning the data in Figure 5/Figure 5—figure supplement 1 and the statements in the second paragraph of the subsection “Helix D1M2 connects KtrB with KtrA”. It was not simply recommended that the discussion be toned down. Therefore, I cannot recommend the publication of this manuscript in eLife in its current form. I would be more than willing to do so if the authors complied with this straightforward request – which is no way detrimental for this study.*

*Reviewer #3:*

*The main conclusions of this paper are based on the cryo EM structure and supporting information from EPR and MD. I think that overall the authors have tried to address the main points and have toned down their conclusions based on the MD analysis. Our suggestion that additional sites be tested in the EPR section proved difficult to obtain, as expected, although the authors have presented data supporting their statement that most residues in the extended helix gave poor or difficult to interpret distance distributions. The authors however stand by their claim that the data qualitatively supports their argument of a structural transition in the D1M2 region following ATP/ADP binding and I would agree with this conclusion. On my specific query regarding the role of lipids, they didn't actually address, instead performing a set of additional experiments in detergent measuring the flexibility in the presence of nucleotide, which does support their arguments, but didn't address the more important question on the role of lipids in stabilizing the D1M2 region. This was never going to be a deal breaker anyway.*

*I am still in favor of this paper being published, as the structural work does present novel insights into the gating mechanism within this K^+^ channel family. The 6 A maps are more than clear enough to propose the jackknife model from a structural perspective. The authors have made suitable changes to the text and data to better support their conclusions, which I think are, on the whole, justified.*

As suggested by reviewer #1 we have now removed the data in previous Figure 5/Figure 5—figure supplement 1 and the statements in the second paragraph of the subsection “Helix D1M2 connects KtrB with KtrA”. In addition, we have adjusted the Materials and methods section accordingly.